# Common elements of service delivery models that optimise quality of life and health service use among older people with advanced progressive conditions: a tertiary systematic review

Joanne Bayly ![ORCID],[1,2] Anna E Bone,[1] Clare Ellis-Smith,[1] India Tunnard,[1] Shuja Yaqub,[1] Deokhee Yi ![ORCID],[1] Kennedy B Nkhoma ![ORCID],[3] Amelia Cook,[1] Sarah Combes,[3,4,5] Sabrina Bajwah,[1] Richard Harding ![ORCID],[1] Caroline Nicholson,[4,5] Charles Normand,[1,6] Shalini Ahuja,[7] Pamela Turrillas,[1] Yoshiyuki Kizawa,[8] Tatsuya Morita,[9] Nanako Nishiyama ![ORCID],[10] Satoru Tsuneto,[11] Paul Ong,[12] Irene J Higginson,[1] Catherine J Evans ![ORCID],[1,13] Matthew Maddocks ![ORCID][1]

CJE and MM are joint senior authors.

For numbered affiliations see end of article.

**Correspondence to**
Dr Matthew Maddocks;
matthew.maddocks@kcl.ac.uk

## ABSTRACT

**Introduction** Health and social care services worldwide need to support ageing populations to live well with advanced progressive conditions while adapting to functional decline and finitude. We aimed to identify and map common elements of effective geriatric and palliative care services and consider their scalability and generalisability to high, middle and low-income countries.

**Methods** Tertiary systematic review (Cochrane Database of Systematic Reviews, CINAHL, Embase, January 2000–October 2019) of studies in geriatric or palliative care that demonstrated improved quality of life and/or health service use outcomes among older people with advanced progressive conditions. Using frameworks for health system analysis, service elements were identified. We used a staged, iterative process to develop a 'common components' logic model and consulted experts in geriatric or palliative care from high, middle and low-income countries on its scalability.

**Results** 78 studies (59 geriatric and 19 palliative) spanning all WHO regions were included. Data were available from 17 739 participants. Nearly half the studies recruited patients with heart failure (n=36) and one-third recruited patients with mixed diagnoses (n=26). Common service elements (≥80% of studies) included collaborative working, ongoing assessment, active patient participation, patient/family education and patient self-management. Effective services incorporated patient engagement, patient goal-driven care and the centrality of patient needs. Stakeholders (n=20) emphasised that wider implementation of such services would require access to skilled, multidisciplinary teams with sufficient resource to meet patients' needs. Identified barriers to scalability included the political and societal will to invest in and prioritise palliative and geriatric care for older people, alongside geographical and socioeconomic factors.

**Conclusion** Our logic model combines elements of effective services to achieve optimal quality of life and health service use among older people with advanced progressive conditions. The model transcends current best practice in geriatric and palliative care and applies across the care continuum, from prevention of functional decline to end-of-life care.

**PROSPERO registration number** CRD42020150252.

## STRENGTHS AND LIMITATIONS OF THIS STUDY

⇒ We draw on and synthesise a diverse evidence base of geriatric and palliative care for older people with progressive advanced conditions across the globe.
⇒ The review was conducted by a multidisciplinary and international group representing broad methodological expertise and perspectives.
⇒ Our common components logic model is a recombination of effective service elements. However, we were unable to assert how outcomes may be influenced by different combinations of components and their interactions.
⇒ Our stakeholder consultation identified significant barriers to scalability where country health budgets cannot meet the growing population need, and where multidisciplinary care is not available.

## INTRODUCTION

Globally, more people are living into old age[1] with the largest proportional increase occurring in those 80 years and above.[2 3] By 2050, 80% of older people will live in low-income and middle-income countries (LMICs).[4] The concomitant risks of multimorbidity and/or frailty[5] mean more people experience a trajectory of prolonged and uncertain functional decline. Health and social care needs and their impact on physical functioning are more heterogeneous[1] in older populations,

shaped by multiple interacting factors related to the individual and their environment. These population changes bring new societal challenges related to health and social care policy, spending, workforce and security, regardless of the developmental context.

The WHO Member States' commitment to achieve Universal Health Coverage (UHC) by 2030 provides an opportunity to plan health and social care delivery for the future. Palliative care has recently been included as an essential service that is fundamental to achieving UHC.[6] While prevention remains a priority across the health continuum, a shift in health systems is needed to balance disease-modifying interventions with services where improving quality of life is the main goal of care. In older people with advanced (incurable) and progressive diseases, health systems must align to support the dual priorities of living well while adapting to a gradual decline in function. Access to appropriate care and support is recognised as a basic human right,[7] yet access varies according to socioeconomic and geographical variables.[8 9] Budget constraints require maximum value from the resources used to improve outcomes.[10] The importance of integrated working across services is consistently advocated in global guidance on health service provision for advanced disease[11] and older people.[12]

Our previous meta-review outlined two service delivery models for older people towards the end of life; 'integrated geriatric care' and 'integrated palliative care'.[12] Both showed potential to improve quality of life and patterns of health service use, but with differing emphasis on either function or symptoms and concerns. Our findings underscored the imperative of access to services based on the likelihood of benefit and integration of care using comprehensive assessment, case management and/or collaborative working.[12] However, use of systematic reviews as the unit of analysis prevented a detailed description of service model elements, and suppressed the heterogeneity across the primary studies.

This review aimed to detail service delivery models that optimise quality of life and health services use for older people aged 60 years and over with advanced progressive health conditions. We defined 'advanced' to include disease stage, people described as in their last 1 or 2 years of life or people accessing a service typically used in advanced disease stage, such as nursing home or palliative care. Our objectives were to: (1) identify and map common elements of effective service delivery models within primary studies; (2) outline the similarities and differences across models of geriatric care or palliative care and (3) consider the scalability of effective models, attending to implementation and economic requirements.

## METHOD
### Study design
This review builds on our previous meta-review, where the methods are described in detail.[12] Here, we conducted a

tertiary review of individual empirical studies ('primary studies') from the meta-review.[12] This was conducted in accordance with the Preferred Reporting Items for Systematic Reviews and Meta-Analysis.[13] We then used logic modelling[14] and a stakeholder consultation to support the analysis and interpretation[15] of findings. This study was registered on PROSPERO prior to data extraction.

### Patient and public involvement
Patients and members of the public were not involved in the design, conduct, reporting or dissemination of this research.

### Search strategy
For the purposes of this tertiary review, in October 2019 we updated our original meta-review search to identify systematic reviews that included a meta-analysis that demonstrated overall effectiveness on at least one outcome for quality of life (including symptom burden and function) and/or health service use outcome. The systematic review eligibility criteria and search terms are reported in online supplemental materials 1 and 2. From the eligible systematic reviews, we identified primary studies with evidence of effect on our selected outcomes of quality of life and/or health service use. Inclusion criteria for primary studies comprised: (1) experimental study design; (2) contributed data to meta-analysis and (3) reported a point estimate of effect in the same direction as the meta-analysis. One reviewer (JB) evaluated all systematic reviews and primary studies for eligibility and a second (MM, AB or CE-S) double-screened studies, with inconsistencies resolved by consensus. Duplicate primary studies were identified and removed.

### Data extraction
Data on study population, outcomes and context were extracted. Service delivery models were classified as either integrated geriatric or palliative care. Data identification and extraction was informed by a framework for healthcare systems analysis, the checklist CATWOE (customers, actors, transformation processes, world view, owner, environmental constraints).[16 17] For each CATWOE domain, (eg, customers, actors), a list of service elements was identified. Service elements were categorised as present, absent or unclear by two individuals (from JB, AB, CE-S, SY, DY, NK, SB, CE, MM) and reviewed as a team. Identification of the elements for each CATWOE domain was, informed by the Template for Intervention Description and Replication checklist for complex health service interventions[18] and prior studies on geriatric,[19] integrated,[20] transitional[21] and palliative care[22] online supplemental material 3 details the elements for each CATWOE domain.

### Quality appraisal
The methodological quality of systematic reviews and primary studies was appraised using the A MeaSurement Tool to Assess systematic Reviews (AMSTAR) tool[23]

and Cochrane Risk of Bias Tool, respectively.[24] We used the quality appraisal in the systematic reviews when the Cochrane Risk of Bias Tool was used, otherwise assessment was by two researchers (JB and IT). We did not exclude studies from analysis based on quality.

### Development of logic model

We used a staged and iterative approach following Rohwer *et al*'s guidance on logic models for complex health interventions[14] incorporating analysis of extracted data followed by a stakeholder consultation.

The frequency and proportion of service elements[16 17] was summarised overall and separately for integrated geriatric and palliative care models. The proportion was calculated using studies where the element was categorised as present or absent. We mapped service elements present in ≥50% of integrated geriatric and/or palliative care studies by CATWOE domain to existing logic model templates[14] (see online supplemental material 4). To compare the presence of service elements between integrated geriatric and palliative care models we conducted $\chi^2$ tests (or Fisher's exact tests where counts were low).

We appraised the potential for the common components of effective interventions to be generalised and scalable, defined as the ability 'to be expanded under real world conditions to reach a greater proportion of the eligible population while retaining effectiveness'.[25]

We shared an interim logic model and consulted a purposive sample of healthcare researchers, clinical-academics and clinicians from high, middle and low-income countries with expertise in either geriatric or palliative care, hospital or community based. We used the Context and Implementation of Complex Interventions Framework (CICI) to develop a response form with free-text open questions on the barriers and facilitators to providing the elements of care, for their respective country and healthcare settings.[26] The CICI framework domains informed the identification and collation of the narrative responses on the context and implementation considerations. We developed the logic model by synthesising the findings from the tertiary review and the stakeholder consultation, using an iterative process of team discussion and consensus.[14]

## RESULTS

### Study retrieval

Ten systematic reviews met eligibility, seven from the meta-review[27–33] and three from the updated search.[34–36] The reviews reported 180 potentially eligible studies, of which 47 were duplicates. Of the 133 remaining studies, 78 met eligibility (figure 1).

### Characteristics of included studies

Of the 78 included studies, 59 were categorised as integrated geriatric care and 19 as integrated palliative care (table 1 and online supplemental material 5). All WHO regions were represented, though studies were

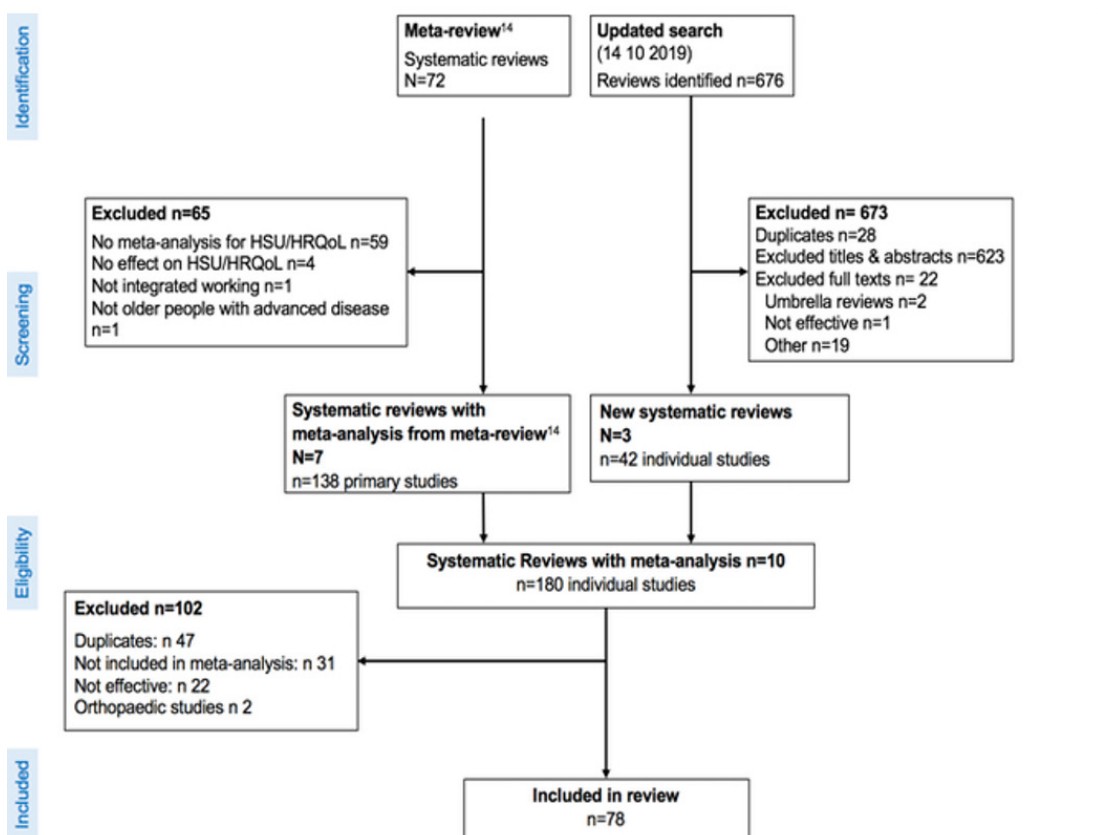

**Figure 1** Preferred Reporting Items for Systematic Reviews and Meta-Analyses flowchart for selection of primary studies.

**Table 1** Summary characteristics of included studies N=78

| Variable | | Frequency n (%) | | |
|---|---|---|---|---|
| | | All n=78 | Geriatric n=59 | Palliative n=19 |
| WHO region | Americas | 46 (59) | 36 (61) | 10 (53) |
| | Europe | 22 (28) | 16 (27) | 6 (32) |
| | Southeast Asia | 3 (4) | 2 (3) | 1 (5) |
| | West Pacific | 6 (8) | 5 (8) | 1 (5) |
| | Africa | 1 (1) | 1 (2) | 0 |
| Country income status | High | 75 (96) | 17 (29) | 58 |
| | Upper—middle | 2 (3) | 1 (2) | 1 |
| | Lower—middle | 1 (1) | 0 | 1 |
| | Low | 0 | 0 | 0 |
| Population by main diagnosis | Heart failure | 36 (46) | 32 (54) | 4 (21) |
| | No main diagnosis | 23 (29) | 23 (39) | 0 |
| | Cancer | 14 (18) | 2 (3) | 12 (63) |
| | Single | 4 (5) | 1 (2) | 3 (16) |
| | Mixed | 10 (13) | 1 (2) | 9 (47) |
| | Heart failure +diabetes | 1 (1) | 1 (2) | 0 |
| | Heart failure +depression | 1 (1) | 1 (2) | 0 |
| | Multiple sclerosis | 1 (1) | 0 | 1 (5) |
| | Mixed diagnosis (COPD, cancer, HF, ILD, MND) | 1 (1) | 0 | 1 (5) |
| | HIV infection | 1 (1) | 0 | 1 (5) |
| Population by referral criteria | People with heart failure | 38 (49) | 34 (58) | 4 (21) |
| | People with acute episode of illness | 17 (22) | 17 (29) | 0 |
| | People with advanced cancer | 13 (17) | 2 (3) | 11(58) |
| | Older people (varied age ranges) | 6 (8) | 6 (10) | 0 |
| | People with HIV | 1 (1) | 0 | 1 (5) |
| | People with multiple sclerosis | 1 (1) | 0 | 1 (5) |
| | Advanced mixed diagnoses | 1 (1) | 0 | 1 (5) |
| | People with cancer commencing chemotherapy | 1 (1) | 1 (2) | 1 (5) |
| Health organisation funding | State funded health organisation | 35 (45) | 26 (44) | 9 (47) |
| | For profit health organisation | 37 (47) | 28 (47) | 9 (47) |
| | Non-profit health organisation | 6 (8) | 5 (8) | 1 (5) |
| Care setting | Mixed settings | 29 (37) | 20 (34) | 9 (47) |
| | Hospital inpatients and home | 6 (8) | 6 (10) | 0 |
| | Hospital inpatients and outpatients | 5 (6) | 5 (8) | 0 |
| | Hospital outpatients and home | 10 (13) | 4 (7) | 6 (32) |
| | Hospital inpatients, outpatients and home | 7 (9) | 4 (7) | 3 (16) |
| | Hospital emergency room and home | 1 (1) | 1 (2) | 0 |
| | Home | 16 (21) | 13 (22) | 3 (16) |
| | Hospital outpatients | 15 (19) | 9 (15) | 6 (32) |
| | Hospital inpatients | 13 (17) | 12 (20) | 1 (5) |
| | Community settings | 3 (4) | 3 (5) | 0 |

COPD, chronic obstructive pulmonary disease; HF, heart failure; ILD, interstitial lung disease; MND, motor neuron disease.

predominantly from the North American region of the Americas (n=46) or Europe (n=22), with fewer from Western Pacific (n=6), Southeast Asia (n=3) and only a single study from Africa. Most studies were from high-income countries (n=75). The number of study participants ranged from 20 to 1632, with data available from

17 739 participants overall. Nearly half of all studies recruited patients with heart failure (n=36) and one-third recruited patients with mixed diagnoses (n=26). Palliative care studies most often recruited by cancer diagnosis (n=12). Study interventions were delivered across multiple care settings (n=31), in participants' homes (n=15) or in hospital (outpatients n=14; inpatients n=12) (table 1).

### Quality appraisal

The 10 systematic reviews were assessed as of moderate quality (online supplemental material 6). Primary studies were assessed as low-to-moderate risk of bias overall (online supplemental material 6). Where high risk of bias was found, this most frequently related to challenges of blinding participants and personnel leading to possible performance and detection bias. Risk of bias tended to be lower for palliative care compared with geriatric care studies (online supplemental material 7).

### Service delivery elements

Services most frequently used collaborative working and case management to support integrated working between professionals (table 2). Patient/family education was present in all studies. Other common elements, present in ≥80% of studies were ongoing assessment, active patient participation and evidence of patient engagement in their care. The least common elements overall were: bereavement support; 24-hour home visits or access to physicians; links to residential hospice facilities and joint provision of care across health and social care services. No studies reported delivering interventions in residential care/nursing homes or use of volunteers.

Comparing between integrated geriatric and palliative care, palliative care interventions had a higher frequency of end-of-life expertise and training, professional psychosocial support, spiritual support and physician home visits. In contrast geriatric care interventions more often featured early rehabilitation assessment and self-management, though the differences were not statistically significant (table 2).

### Service delivery agents

All interventions were delivered by qualified healthcare professionals (staff who have received nationally recognised and regulated training and education), most often working in multidisciplinary teams. Over 90% of studies involved trained medical and nursing clinicians and 59% involved members of the wider healthcare team, including physiotherapists, occupational therapists and social workers. Geriatric care services were delivered by physicians from geriatrics, cardiology and general practice, whereas palliative care services involved physicians from cardiology, neurology, respiratory medicine, oncology, psychiatry, primary care and palliative medicine. While involvement of physiotherapists was reported across all studies, fewer occupational therapists and dietitians were reported in those from palliative care. No studies reported the involvement of volunteers (table 3).

### Service outcomes including costs

Forty-five studies (58%) were included based on an effect on quality of life alone. Fifty-seven studies (73%) used a disease or population specific measure to quantify quality of life (online supplemental material 5) and five studies (6%) employed the EuroQoL 5-Dimension (EQ-5D). Thirty-three studies (42%) reported utilisation of acute care services (eg, hospital admission, readmission after discharge) or community care services and 20 studies (26%) calculated costs of health services utilisation. Only a minority (n=12%/15%) demonstrated an effect on both quality of life and health service use, all of which were geriatric care studies. No study used costs and EQ-5D to generate information required for health economic decision-making (table 4).

### Common components logic model

The interim logic model highlighted key elements present in the majority (<80%) of included studies.

Elements more common in integrated palliative care compared with geriatric care studies were; professional psychosocial support, advance care planning, caregiver engagement, joint decision-making and expert consultation with other providers. Elements more common in geriatric care studies included a social worker or dietician as a delivery agent and care planning organised around the service, delivering the same intervention to all patients but with individual tailoring (figure 2).

Elements more common in geriatric care studies included a social worker or dietician as a delivery agent and care planning organised around the service with the same intervention being delivered to all patients with individual tailoring (figure 2).

### Stakeholder perspectives on scalability

The context and implementation considerations identified from the stakeholder responses were incorporated into the logic model (figure 2). Stakeholders (n=20) contributed views from high-income countries (n=12) (UK, Japan, Taiwan, Portugal and Chile) and LMICs (n=8) (Uganda, Malawi, South Africa, Ghana, Zimbabwe, China, India and Bangladesh) contributed views. They described increasing patient complexity with rapid population ageing and the associated rise in multimorbidity, frailty and dementia. This raised particular challenges in LMICs where health services have historically focused on prevention and management of infectious diseases and where there has been a recent increased burden of noncommunicable disease. Specialist services being based in major city hospitals were described as a barrier to providing care to rural populations. Recruiting, training and retaining skilled staff to work in rural areas and having a multidisciplinary team including allied health professionals and specialist doctors and nurses was considered infeasible for many rural areas.

**Table 2** Service delivery model elements N=78

| | All n (%) | Geriatric n (%) | Palliative n (%) | Sig† |
|---|---|---|---|---|
| *Method of supporting integrated working* | | | | |
| Collaborative working | 64 (82) | 46 (78) | 18 (95) | 0.17* |
| Case management | 61 (78) | 46 (78) | 15 (79) | 1.00* |
| Comprehensive assessment | 51 (65) | 36 (68) | 15 (79) | 0.36 |
| *Actors-workforce* | | | | |
| Professional education | 76 (100) | 58 (100) | 18 (100) | 1.00 |
| MDT care | 54 (72) | 42 (73) | 12 (71) | 1.00* |
| Rehabilitation expertise training | 34 (50) | 27 (50) | 7 (50) | 1.00 |
| End-of-life expertise training | 18 (25) | 1 (2) | 17 (90) | <0.001* |
| *Transformation-service model elements/components* | | | | |
| Patient family education | 60 (100) | 49 (100) | 11 (100) | 0.93 |
| Medication review | 51 (80) | 40 (77) | 11 (92) | 0.43* |
| Self-management | 48 (80) | 41 (84) | 7 (64) | 0.21* |
| Systematic risk screening | 47 (69) | 37 (70) | 10 (67) | 1.00* |
| Contact with GP or attending doctor | 46 (68) | 33 (65) | 13 (77) | 0.37 |
| Practical support | 41 (68) | 34 (69) | 7 (64) | 0.73* |
| Medical intervention | 52 (67) | 39 (66) | 13 (68) | 0.85 |
| Individualised MDT plan | 40 (61) | 29 (59) | 11 (65) | 0.69 |
| Complex/medication management | 37 (58) | 30 (59) | 7 (54) | 0.75 |
| Discharge planning | 36 (52) | 29 (55) | 7 (44) | 0.44 |
| Professional psychosocial support | 38 (51) | 26 (44) | 12 (80) | 0.01 |
| Team case rounds | 25 (40) | 18 (37) | 7 (50) | 0.37 |
| Early rehab assessment | 25 (38) | 21 (40) | 4 (29) | 0.54 |
| Advanced care planning | 23 (30) | 9 (16) | 14 (78) | <0.001 |
| Emergency response plan | 15 (21) | 12 (22) | 3 (20) | 1.00* |
| Spiritual support | 13 (18) | 2 (3) | 11 (79) | <0.001* |
| Bereavement support | 4 (5) | 0 (0) | 4 (25) | 0.002* |
| *Transformation-mode of delivery* | | | | |
| Ongoing assessment | 66 (87) | 50 (86) | 16 (89) | 1.00* |
| Face-to-face and telephone | 41 (53) | 31 (53) | 10 (53) | 0.10 |
| Face-to-face interaction | 31 (40) | 23 (39) | 8 (42) | 0.81 |
| Access to inpatient beds | 21 (30) | 18 (32) | 3 (21) | 0.53* |
| Physician home visits | 11 (15) | 4 (7) | 7 (37) | 0.04* |
| 24-hour physician access | 6 (10) | 5 (11) | 1 (7) | 1.00* |
| Telephone only | 5 (6) | 4 (7) | 1 (5) | 1.00* |
| 24-hour home visits | 1 (1) | 1 (2) | 0 (0) | 1.00* |
| Online only | 1 (1) | 1 (2) | 0 (0) | 0.10* |
| *Transformation-operational tools and guidance to support practice* | | | | |
| Standard comprehensive assessment | 38 (59) | 26 (55) | 12 (71) | 0.27 |
| *Worldview-methods of integrated working* | | | | |
| Link to hospital | 57 (78) | 41 (72) | 16 (100) | 0.02* |
| Expert consult with other providers | 40 (58) | 24 (45) | 16 (100) | <0.001 |
| Link between community services | 31 (50) | 22 (45) | 9 (69) | 0.12 |
| Joint provision-health and social care | 7 (10) | 4 (7) | 3 (20) | 0.16* |
| Link to residential hospice | 5 (7) | 1 (2) | 4 (27) | 0.005* |

**Table 2**  Continued

|  | All n (%) | Geriatric n (%) | Palliative n (%) | Sig† |
|---|---|---|---|---|
| *Worldview-conceptual model* | | | | |
| Patient engagement | 71 (99) | 53 (98) | 18 (100) | 1.00* |
| Active patient participation | 67 (99) | 50 (98) | 17 (100) | 1.00* |
| Centrality of patient needs | 64 (91) | 46 (89) | 18 (100) | 0.33* |
| Patient goal driven care | 56 (81) | 40 (77) | 16 (94) | 0.16* |
| Ongoing/continuous care | 46 (67) | 33 (62) | 13 (81) | 0.16 |
| Joint decision-making | 38 (69) | 25 (61) | 13 (93) | 0.04* |
| Service driven care planning | 38 (54) | 34 (65) | 4 (21) | 0.001* |
| Needs and benefit-driven care planning | 33 (46) | 18 (35) | 15 (79) | 0.001 |
| Caregiver engagement | 32 (55) | 22 (50) | 10 (71) | 0.16 |

*Fisher's exact test.
†Sig=significance for difference in presence of service delivery element between geriatric and palliative care studies.
GP, General Practitioner; MDT, Multidisciplinary Team.

Stakeholders from LMICs considered that overall health budgets were inadequate to meet the population need, and multidisciplinary care was considered unaffordable. The voluntary sector was often seen as important to augment publicly funded services. In some contexts, continuity of care is impeded when individually funded services compete for resources rather than collaborate. There are challenges to multidisciplinary working in systems where health workers receive payment directly from patients, as this was considered a financial disincentive to making referrals for expert consultation. Social deprivation was cited as an important barrier to accessing care, especially in health systems with out-of-pocket expenses or private insurance.

Stakeholders described how cultural norms influence care provision. Death denying attitudes in some cultures influence uptake of palliative care services. Some countries have limited recognition or respect for the specialties of palliative care and geriatric care. The role of the family and the health system to provide care was identified to vary across countries influenced by cultural beliefs such as filial piety, gender-related norms and changing intergenerational family structures. Acknowledging faith and religion were identified as factors supporting the delivery of individualised care aligned with spiritual needs in hospice and nursing homes.

Increasing education levels and internet access were identified as factors that are changing patient and family participation in joint decision-making. Finally, stakeholders recognised an increasing political will to invest in services for older people supported by a growing public and research agenda and established regulatory frameworks. However, this did not always equate to increases in funding. A lack of policies and clinical governance for specialist palliative and geriatric care was reported as a problem, like legal restrictions on opioid prescribing limiting effective medication management of pain.

## DISCUSSION

We used rigorous methods to detail service delivery models that optimise quality of life and health service use outcomes among older people with advanced progressive conditions. Effective services commonly used collaborative working between professionals and specialties, comprehensive and ongoing assessment, patient/family education and active patient participation. Aligned to this, effective services consistently incorporated patient engagement, patient goal-driven care and the centrality of patient needs in care delivery. Our logic model encompasses a breadth of elements that aim to 'protect' (discharge planning and falls prevention programmes), 'reactivate' (disease management, self-management and exercise programmes), 'compensate' (symptom management, support with capabilities for activities of daily living) and 'support' (enhancing social assets and provision of home care). Such practices may together support older people to maintain intrinsic capacity and functional ability[37] and to compress functional decline across the life course.[38 39] This broad focus, together with consideration of social factors, extends health and social care beyond provision at the point of decline to meet the dual priorities of living well while adapting to a gradual functional decline.[1]

This review has several strengths. It was conducted by a large multidisciplinary team with a range of methodological expertise and representation from many regions of the world. We synthesised a diverse literature with studies across different patient populations and needs across the trajectory of advanced disease. We used recognised frameworks to categorise studies, extract data and consult with stakeholders in order to develop a visual logic model applicable to different international settings. There are some limitations to consider. Data on study context is limited to country, country income status and the system for funding healthcare. Further information to support

**Table 3** Service delivery model agents

| Delivery agent | All n (%) | Geriatric n (%) | Palliative n (%) | Sig† |
|---|---|---|---|---|
| *Physicians* | | | | |
| Geriatrician | 14 (18) | 14 (24) | 0 (0) | 0.02 |
| Cardiologist | 15 (19) | 12 (20) | 3 (16) | 1.0 |
| Palliative care physician | 12 (15) | 0 (0) | 12 (63) | <0.001* |
| Neurologist | 1 (1) | 0 (0) | 1 (5) | 0.24* |
| Respiratory physician | 1 (1) | 0 (0) | 1 (5) | 0.24* |
| Oncologist | 4 (5) | 0 (0) | 4 (21) | 0.001* |
| Psychiatrist | 2 (3) | 0 (0) | 2 (11) | 0.06* |
| Physician | 18 (23) | 17 (29) | 1 (5) | 0.06* |
| Primary care doctor (GP) | 5 (6) | 4 (7) | 1 (5) | 0.55* |
| Physician assistant | 2 (3) | 2 (3) | 0 (0) | 0.43* |
| *Nurses* | | | | |
| Nurse | 24 (31) | 22 (37) | 2 (11) | 0.28 |
| Advanced nurse practitioner | 13 (17) | 8 (14) | 5 (26) | 0.17* |
| Specialist cardiac nurse | 12 (15) | 10 (17) | 2 (11) | 0.40* |
| Primary care nurse | 9 (8) | 8 (14) | 1 (5) | 0.30* |
| Specialist geriatric nurse | 6 (8) | 6 (10) | 0 (0) | 0.18* |
| Case manager | 5 (6) | 3 (5) | 2 (11) | 0.35* |
| Specialist palliative nurse | 4 (5) | 1 (2) | 3 (16) | 0.43* |
| Specialist rehabilitation nurse | 1 (1) | 1 (2) | 0 (0) | 0.76* |
| Specialist HIV nurse | 1 (1) | 0 (0) | 1 (5) | 0.24* |
| Oncology nurse | 1 (1) | 0 (0) | 1 (5) | 0.24* |
| *Allied health professionals* | | | | |
| Physiotherapist | 23 (29) | 17 (29) | 6 (32) | 0.85 |
| Occupational therapist | 14 (18) | 12 (20) | 2 (11) | 0.28* |
| Dietitian | 16 (21) | 14 (24) | 2 (11) | 0.18* |
| Psychologist | 9 (15) | 6 (10) | 3 (16) | 0.38* |
| Pharmacologist/pharmacist | 7 (9) | 7 (12) | 0 (0) | 0.13* |
| Chaplain | 4 (5) | 1 (2) | 3 (16) | 0.43* |
| Audiologist | 1 (1) | 1 (2) | 0 (0) | 0.76* |
| Speech and language therapist | 1 (1) | 1 (2) | 0 (0) | 0.76* |
| *Social care* | | | | |
| Social worker | 21 (27) | 17 (29) | 4 (21) | 0.51 |
| Home care service manager | 3 (4) | 3 (5) | 0 (0) | 0.43* |
| Social assistant | 4 (1) | 3 (5) | 1 (5) | 0.68* |
| *Other professionals* | | | | |
| Unspecified wider 'MDT' | 11 (14) | 9 (15) | 2 (11) | 0.47* |
| Exercise instructor | 2 (3) | 2 (3) | 0 (0) | 0.57* |

*Fisher's exact test.
†Significance for difference in presence of service delivery element between geriatric and palliative care studies.

an evaluation of how interventions could be scaled and implemented would be valuable. Stakeholders identified limited applicability for some service elements, including multidisciplinary care, within low-income countries where health budgets cannot meet the growing population need. Change beyond the health system, into education and health promotion, may be required to implement services that meet the challenge of rising incidence in diseases of ageing.[40] As in other reviews of complex interventions in this population,[16] we were

**Table 4** Number of studies reporting quality of life and health services use outcomes

| | | Health service use | | | |
|---|---|---|---|---|---|
| | | None | More than 1 | 1+and costs | Subtotal |
| Quality of life | None | 0 | 6 | 15 | 21 |
| | More than 1 | 40 | 7 | 5 | 52 |
| | 1+ and EQ-5D | 4 | 0 | 0 | 5 |
| | Subtotal | 45 | 13 | 20 | 78 |

EQ-5D, EuroQoL 5-Dimension.

unable to discern the specific mechanisms of action that make each component effective. In part this was linked to our data extraction framework. For example, we did not extract data on how interventions provided care across care boundaries during care transitions, yet elements including ongoing assessment and links between community services indicate this may have been occurring.

Our findings build on previous reviews. Bainbridge et al[22] found that 'linkages with hospital,' 'multiprofessional teams' and 'end of life care expertise and training' were critical to the delivery of end-of-life home care. In a review of integrated care for older people, Briggs et al[20] found that multidisciplinary teams, comprehensive assessment and case management were most frequently reported. We

show the importance of a capable workforce that works collaboratively across disciplinary boundaries, to provide comprehensive and ongoing multidimensional assessment. This model of care requires active patient engagement, participation and self-management with tailored care centred on the needs of individuals.[41] It allows for a shared understanding between the person(when able and/or the family) and the team providing their care, facilitating joint decision-making that addresses their priorities in their context.[42]

We also provide new insights into the range of health and social care providers associated with effective services in this population. Services were frequently delivered by multidisciplinary teams of health and

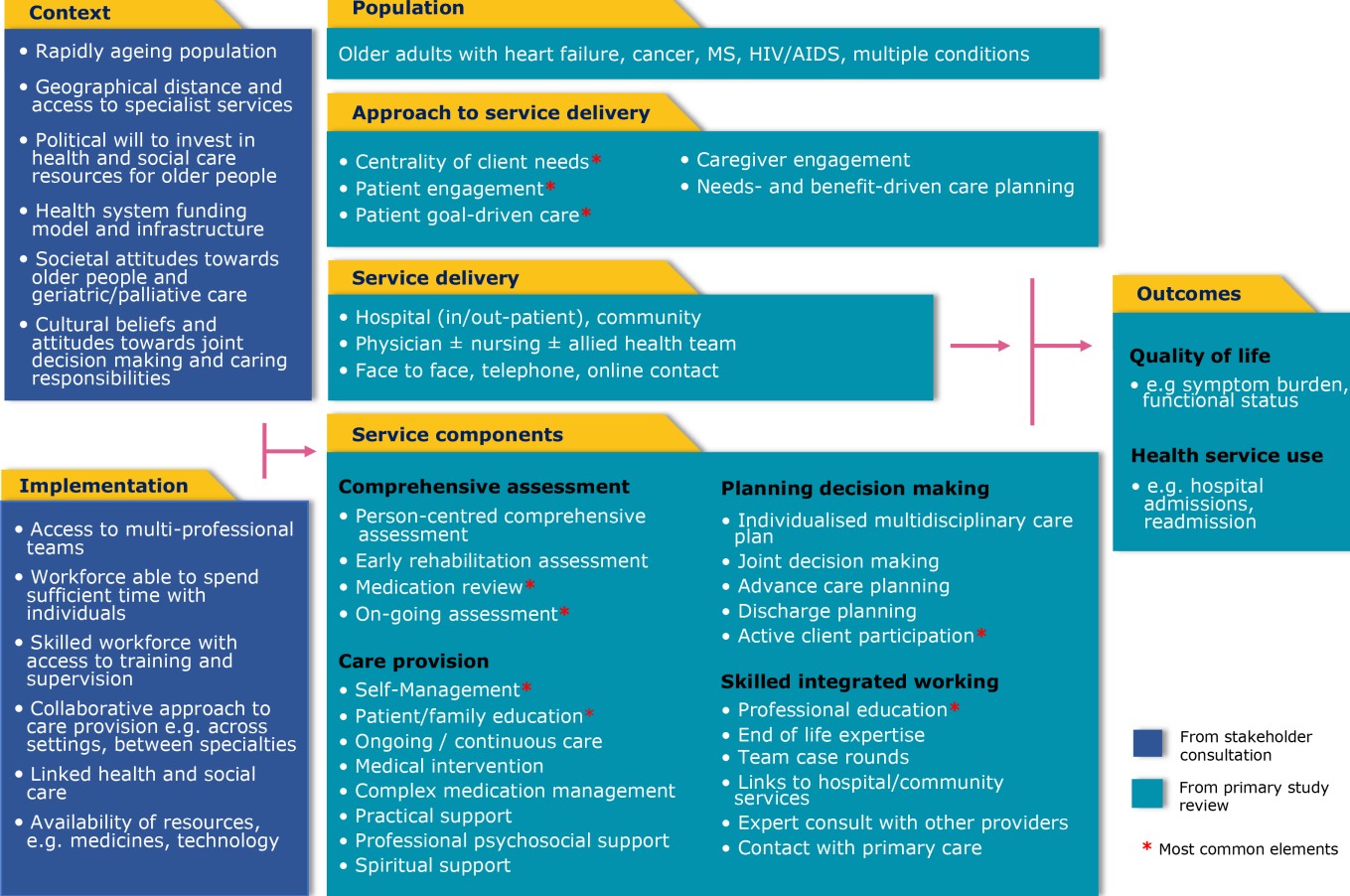

**Figure 2** Common components logic model detailing effective service delivery models for older people with advanced progressive conditions .

social care professionals with formal training in core skills of comprehensive assessment, communication and symptom management. These teams can support people to self-manage a progressive condition and help people close to them to provide care. Investment in training and education is required to achieve greater coverage and ensure the skills base keeps up with the needs of this growing population. Uncoupling skills from professional roles and working towards a generalist skills set may be most beneficial. However, this should ideally be accompanied by access to specialists for ongoing support and supervision. Volunteers may provide additional support that supplements or enhances usual health and social care provision.[43 44] The absence of volunteers in studies probably reflects the fact that most were conducted in high-income countries.

Service elements that we consider relevant to the target population but not brought forward to our logic model include joint provision across health and social care and early rehabilitation assessment. Neglecting social care can have a considerable negative effect on quality of life for older people, their family and friends and lead to increased patient and carer morbidity and mortality.[45] Integrated care should follow older people as they transition from acute to community care.[46] However workforce issues continue to influence the integration of health and social care delivery[47] as highlighted in our stakeholder consultation. Early rehabilitation assessment was detailed in only 40% and 19% of geriatric and palliative care studies, respectively. Given that maintaining independence, normality and social participation are high priorities for older people towards the end of life,[48] this was a surprising finding. It may relate to a focus on physical symptoms arising from advanced disease rather than functional needs, and the presumption that decline is an inevitability of disease progression.[49] The increasing prominence of rehabilitation in palliative care to challenge this misconception is therefore timely.[50 51]

The logic model is a recombination of different services and we were unable to assert how effectiveness may be influenced by different combinations of components and their interactions. Consequently, the model remains untested as a whole.[52] However the model can inform health and social care policy and support the conceptual and organisational development of services. We recommend that the clinical and cost-effectiveness of interventions, underpinned by our proposed model, should be tested in older people with multimorbidity based on need, rather than diagnostic condition, over longer trajectories and across care boundaries. Implications for policy are presented in box 1.

## CONCLUSION

Our logic model brings together common elements of interventions found to optimise quality of life and health service use among older people with advanced progressive conditions. These included collaborative working

---

### Box 1    Implications for policy

⇒ Configure services for the whole trajectory of chronic progressive conditions up until the end of life and move away from a focus on acute episodes of care.
⇒ Plan and deliver education to drive provision of a capable workforce. A broad range of professional education courses and training in core skills of geriatric and palliative care, including comprehensive assessment, communication and symptom management specific to individual need is required.
⇒ Incentivise interdisciplinary and collaborative working between professional disciplines and across health and social care settings, to optimise high-quality individualised service provision and care co-ordination. This integrated care, when aligned to need rather than diagnostic condition, will increase the reach and impact of services and promote equitable access.
⇒ Enable robust evaluation by embedding routine outcome measurement in health and social care settings. These should include measures of intrinsic capacity, functional ability, symptom experience and quality of life. Measures should capture the changes in health and social well-being that are associated with the provision of high quality individualised care across the care continuum from protect to support and end-of-life care.

---

between professionals and specialties, ongoing assessment, active patient participation, patient/family education and patient self-management, while effective service delivery approaches consistently incorporated patient engagement, patient goal-driven care and the centrality of patient needs.

These elements transcend best practices in geriatric care and palliative care to optimise patient outcomes across the continuum, from prevention of functional decline to end-of-life care. The logic model serves as a useful resource for health systems looking to strengthen their response to population ageing. It can guide provision of health and social care that is aligned to the needs of this rapidly growing population. Such care should allow older people across the globe to live fully, with minimal suffering, and to die with dignity.

**Author affiliations**
[1]Cicely Saunders Institute for Palliative Care, Policy and Rehabilitation, King's College London, London, UK
[2]St Barnabas Hospice, Worthing, UK
[3]Florence Nightingale Faculty of Nursing Midwifery and Palliative Care, King's College London, London, UK
[4]St Christopher's Hospice, London, UK
[5]University of Surrey Faculty of Health and Medical Sciences, Guildford, UK
[6]Centre for Health Policy and Management, The University of Dublin Trinity College, Dublin, Ireland
[7]Health Service and Population Research Department, King's College London Institute of Psychiatry Psychology and Neuroscience, London, UK
[8]Department of Palliative Medicine, Kobe University, Kobe, Japan
[9]Palliative and Supportive Care Division, Seirei Mikatahara Hospital, Hamamatsu, Japan
[10]Graduate School of Comprehensive Rehabilitation, Osaka Prefecture University, Habikino, Japan
[11]Department of Human Health Sciences, Department of Palliative Medicine, Kyoto University Hospital, Kyoto, Japan
[12]WHO Centre for Health Development (WKC), Kobe, Japan
[13]Sussex Community NHS Foundation Trust, Brighton, UK

**Acknowledgements** We thank Olivia Dix for proofreading the manuscript and Hamid Benalia for helping with the production of the figures.

**Contributors** JB, AB, CE-S, RH, CNo, SA, PT, YK, TM, NN, ST, PO, IH, CE and MM conceived and designed the study. JB, AB, CE-S, IT, SY, DY, KBN, AC, SC, SB, CE and MM extracted data. JB, AB, CE-S, SY, AC, SC, CNi, CNo, RH, KBN, CE and MM analysed data. JB, AB, AC, SC, CNi, PO, CE and MM drafted the manuscript, All authors critically revised the draft and approved the final manuscript. MM is responsible for the overall content as the guarantor.

**Funding** This research was supported by the World Health Organization Centre for Health Development (WHO Kobe Centre – WKC: K19002), the Dunhill Medical Trust (grant number RPGF1906\177) and the National Institute of Health Research Applied Research Collaboration South London (NIHRARC South London NIHR200152) at King's College Hospital NHS Foundation Trust. AB is supported by the Dunhill Medical Trust (number RPGF1906\177) and Cicely Saunders International. SC is funded by a Health Education England/NIHR Clinical Doctoral Research Fellowship (ICA-CDRF-2017-03-012) and CNi by a Health Education England/NIHR Senior Clinical Lectureship (ICA-SCL-2018-04-ST2-001). IH is an NIHR Senior Investigator Emeritus. CE is funded by a Health Education England/NIHR Senior Clinical Lectureship (ICA-SCL-2015-01-001) and MM is funded by an NIHR Career Development Fellowship (CDF-2017-10-009). This publication presents independent research supported by the National Institute for Health Research (NIHR). The views expressed in this publication are those of the author(s) and not necessarily those of the NHS, NIHR or the Department of Health and Social Care.

**Competing interests** PO reported that he was an employee of the funding sponsor, the WHO, and was involved in the extraction, analysis and interpretation of data. All other authors have no competing interests.

**Patient consent for publication** Not applicable.

**Provenance and peer review** Not commissioned; externally peer reviewed.

**Data availability statement** Data are available upon reasonable request. Extracted data is available on request from the corresponding author..

**ORCID iDs**
Joanne Bayly http://orcid.org/0000-0001-9478-8932
Deokhee Yi http://orcid.org/0000-0003-4894-1689
Kennedy B Nkhoma http://orcid.org/0000-0002-2991-8160
Richard Harding http://orcid.org/0000-0001-9653-8689
Nanako Nishiyama http://orcid.org/0000-0002-7580-4009
Catherine J Evans http://orcid.org/0000-0003-0034-7402
Matthew Maddocks http://orcid.org/0000-0002-0189-0952

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
