## [Reviewer comments · BMJ Open]

ARTICLE DETAILS

TITLE (PROVISIONAL)	Common elements of service delivery models that optimise quality of life and health service use among older people with advanced progressive conditions: a tertiary systematic review
AUTHORS	Bayly, Joanne; Bone, Anna; Ellis-Smith, Clare; Tunnard, India; Yaqub, Shuja; Yi, Deokhee; Bashan Nkhoma, Kennedy; Cook, Amelia; Combes, Sarah; Bajwah, Sabrina; Harding, Richard; Nicholson, Caroline; Normand, Charles; Ahuja, Shalini; Turrillas, Pamela; Kizawa, Yoshiyuki; Morita, Tatsuya; Nishiyama, Nanako; Tsuneto, Satoru; Ong, Paul; Higginson, Irene; Evans, Catherine; Maddocks, Matthew

VERSION 1 – REVIEW

REVIEWER	Talevski, Jason University of Melbourne, Medicine (Western Health)
REVIEW RETURNED	16-Mar-2021

GENERAL COMMENTS	The aim of this systematic review is needed in the space of health services research at the moment. The review has sound methodology and the authors conducted the review appropriately. I believe the manuscript should be published, although after major revision. See comments below. Introduction  • Define “oldest old” • “Successes in child and maternal health and infectious diseases pose new challenges for global health” – I don’t understand how a success poses a challenge? • Define “advanced progressive conditions” Methods  • What do you mean by? “primary study” • The authors refer to their previous review a lot. My suggestion is add one sentence at the beginning such as “This review builds on our previous review (ref), in which the methods are described in detail. In brief.....” then go on to briefly explain your methods. • Go into more detail about the chi squared or Fisher’s exact tests. Results  • Table 1 report % as well • Table 2: add a footnote that explains what the significance is for (between the two groups). • Service outcomes including costs: You report what was measured but not the actual results (e.g. were some programs effective at improving QoL?) Discussion
--

	 • Too long! Cut by 20% (at least) Overall comments  • The overall manuscript is quite long and many sentences/paragraphs through the results and discussion section are very repetitive. I suggest cutting unneeded details. • The standard of written English is not acceptable for publication as is. Please review sentence structures and grammar throughout the article.
--	--

REVIEWER	Smith, Kate University of Western Australia
REVIEW RETURNED	29-Mar-2021

GENERAL COMMENTS	Comments to the author This systematic review is an important contribution to improving our understanding on how services may better assist older people to have a good life. The paper gives an overview of service elements which may enhance quality of life and/or health service use in older people, and integrates these into a logic model. This model and subsequent recommendations have the potential to inform better care for older people, with possible limitations to broader translation/scalability discussed by the authors. I have some questions/comments for the authors below.  1. How many databases were used for the search? Supplementary 2 shows the search strategy for Medline, what others were used? 2. What was the date range used for searching publications? 3. Under search strategy, first sentence, add “for eligibility criteria and search terms see Supplementary material 1 and 2 ” to make it easier to locate for the reader. 4. P8, lines 44-52, is very unclear. Can this information regarding mapping of elements from the CATWOE framework names into different categories/headings either be put in a table, or written more clearly? There is an inconsistent use of colons and semi-colons. Figure 2 does not seem to include a heading on “approach to service delivery”. Is this under the service delivery heading of the final logic model, or was this dropped from the interim model following the stakeholder consultation? Where was Worldview mapped, was this under context or underpinning theory? Can you make it clearer as to whether the reader should be referring to the logic model or Table 2 (e.g. see Table 2) for this section. 5. The logic model needs to be enlarged x2 to be able to clearly view the elements. 6. Page 11, Patient family education was in 100% of studies but I could not see it mentioned as a common service delivery element in results, although it is in the discussion. 7. How is living well measured in the 57 studies which had a quality of life outcome (EQ-5D is noted, what were the other measures)? It would be useful to know if the outcome of the primary studies was that patient perspectives of their wellbeing had improved (encompassing worldviews and perspectives of living well), or whether the primary outcome was improvement in patient function or reducing symptom burden (which may or may not have an impact on a person’s perspective of their wellbeing). To make this clearer the outcome measure used for measuring QoL in each study should be included in the Table Supplementary material 4 (where possible). It is a shame there was not more data collected in the studies to inform health economic decisions. 8. Pg 15 line 56 add (n=stakeholders from LMIC)
--

	9. Correct typo on Supplementary material 6 (selective reporting heading). Thank you for the opportunity to read and review your paper.
--	--

VERSION 1 – AUTHOR RESPONSE

Reviewer: 1

Mr. Jason Talevski, University of Melbourne, The University of Melbourne and Western Health

Comments to the Author:

The aim of this systematic review is needed in the space of health services research at the moment. The review has sound methodology and the authors conducted the review appropriately. I believe the manuscript should be published, although after major revision. See comments below.

Thank you for this positive feedback regarding publication. We have amended the paper following consideration of your comments which we believe has improved the manuscript.

Introduction

• Define “oldest old”

We agree it will be useful to clarify this sentence as there are variations in the literature for the 'oldest old'. We provide the figure provided in the United Nations report cited, as below:

'Globally, more people are living into old age [1] with the largest proportional increase occurring in those 80 years and above [2, 3].'

• “Successes in child and maternal health and infectious diseases pose new challenges for global health” – I don’t understand how a success poses a challenge?

We agree this could be clearer, we have amended the sentences to read:

'The concomitant risks of multi-morbidity and/or frailty [5] mean more people experience a trajectory of prolonged and uncertain functional decline. Health and social care needs and their impact on physical functioning are more heterogeneous[1] in older populations, shaped by multiple interacting factors related to the individual and their environment. These population changes bring new societal challenges related to health and social care policy, spending, workforce and security, regardless of the developmental context.'

• Define “advanced progressive conditions”

We have used more precise language to convey the population who are the focus of this review.

'This review aimed to detail service delivery models that optimise quality of life and health services use for older people aged 60 years and over with advanced progressive health conditions. We defined 'advanced' to include disease stage, people described as in their last one or two years of life or people accessing a service typically used in advanced disease stage, such as nursing home or palliative care.'

Methods

• What do you mean by? “primary study”

Thank you for requesting clarification. We understand 'primary study' to be an individual empirical study that can be included in meta-analysis in a systematic review. We have amended the paragraph as can be seen in the amended text following the next comment.

• The authors refer to their previous review a lot. My suggestion is add one sentence at the beginning such as “This review builds on our previous review (ref), in which the methods are described in detail. In brief.....” then go on to briefly explain your methods.

Thank you for these suggestions. To make our methods clearer, we have amended the Study Design paragraph to read as follows:

'This review builds on our previous meta-review, where the methods are described in detail [12]. Here, we conducted a tertiary review of individual empirical studies ('primary studies') from the meta-review [12].'

• Go into more detail about the chi squared or Fisher's exact tests.

We have more fully explained the method for use of Fisher's Exact Test as follows:

'To compare the presence of elements between integrated geriatric and palliative care models we conducted chi squared tests (or Fisher's exact tests where counts were low).'

Results

• Table 1 report % as well

We have added % as suggested

• Table 2: add a footnote that explains what the significance is for (between the two groups).

We have added a footnote to Tables 2 & 3 to explain the significance test.

• Service outcomes including costs: You report what was measured but not the actual results (e.g. were some programs effective at improving QoL?)

Thank you for requesting clarification here. We have not reported details of primary study results as the focus of the review was to identify service components in effective studies

identified from meta-analysis. We have revised the method under the 'search strategy' section to more clearly indicate that we only included systematic reviews with evidence of effect on our selected outcomes, and from those reviews, only primary studies with a point estimate in the direction of effect.

'From these systematic reviews, we identified primary studies with evidence of effect on our selected outcomes. Inclusion criteria for primary studies comprised: i) experimental study design; ii) contributed data to meta-analysis; and iii) reported a point estimate of effect in the same direction as the meta-analysis'.

Discussion

- **Too long! Cut by 20% (at least)**

We acknowledge that the discussion was too long with detail that detracted from the main findings of the review. We have reduced it by approximately 30%.

Overall comments

- **The overall manuscript is quite long and many sentences/paragraphs through the results and discussion section are very repetitive. I suggest cutting unneeded details.**
- **The standard of written English is not acceptable for publication as is. Please review sentence structures and grammar throughout the article.**

We have carefully reviewed the manuscript and have reduced the word count. We reduced unneeded detail and repetition and improved the sentence structures and grammar. We engaged a professional proof reader to review the final manuscript and comment on the editorial style and the reporting. We have detailed this contribution in the acknowledgement.

Reviewer: 2

Dr. Kate Smith, University of Western Australia

Comments to the Author:

Comments to the author

This systematic review is an important contribution to improving our understanding on how services may better assist older people to have a good life. The paper gives an overview of service elements which may enhance quality of life and/or health service use in older people, and integrates these into a logic model. This model and subsequent recommendations have the potential to inform better care for older people, with possible limitations to broader translation/scalability discussed by the authors.

Thank you for this positive feedback.

I have some questions/comments for the authors below.

1. How many databases were used for the search? Supplementary 2 shows the search strategy for Medline, what others were used?
2. What was the date range used for searching publications?

We have added detail to Supplementary Material 2 to indicate which databases were searched and the date range.

"Supplementary material 2. Search Strategy for Medline

'The search strategy was adapted for searches on The Cochrane Database of Systematic Reviews, CINAHL and Embase databases [14] and included studies published between January 2000 and October 2017.'

3. Under search strategy, first sentence, add “for eligibility criteria and search terms see Supplementary material 1 and 2” to make it easier to locate for the reader.

Thank you for your suggestions to improve the clarity of the Search strategy paragraph. We have amended the paragraph as follows:

'For the purposes of this tertiary review, in October 2019 we updated our original meta-review search to identify systematic reviews that included a meta-analysis that demonstrated overall effectiveness on at least one outcome for quality of life (including symptom burden and function) and/or health service use outcome. The systematic review eligibility criteria and search terms are reported in Supplementary materials 1 and 2. From the eligible systematic reviews, we identified primary studies with evidence of effect on our selected outcomes of quality of life and/or health service use. Inclusion criteria for primary studies comprised: i) experimental study design; ii) contributed data to meta-analysis and iii) reported a point estimate of effect in the same direction as the meta-analysis.'

4. P8, lines 44-52, is very unclear. Can this information regarding mapping of elements from the CATWOE framework names into different categories/headings either be put in a table, or written more clearly? There is an inconsistent use of colons and semi-colons. Figure 2 does not seem to include a heading on “approach to service delivery”. Is this under the service delivery heading of the final logic model, or was this dropped from the interim model following the stakeholder consultation? Where was Worldview mapped, was this under context or underpinning theory? Can you make it clearer as to whether the reader should be referring to the logic model or Table 2 (e.g. see Table 2) for this section.

Thank you for requesting this clarification. We agree this paragraph needs to be clearer. We have followed your suggestion and put the mapping process between the CATWOE domains and the Logic Model template domains in a figure which can be found in Supplementary Material 4. This figure in Supplementary Material 4, clarifies how worldview elements are mapped to the logic model template. We have amended the text in the main manuscript as follows:

'We mapped service elements present in ≥50% of integrated geriatric and/or palliative care studies by CATWOE domain to existing logic model templates [14] (see Supplementary Material 4).'

It appears an earlier version of Figure 2 was submitted. The amended figure does contain the heading 'Approach to Service Delivery'.

5. The logic model needs to be enlarged x2 to be able to clearly view the elements.

We have redesigned figure 2 to make the elements more clearly visible.

6. Page 11, Patient family education was in 100% of studies but I could not see it mentioned as a common service delivery element in results, although it is in the discussion.

Thank you for highlighting this. We have amended the results text to include this as below:

'Patient/family education was present in all studies. Other common elements, present in ≥80% of studies were on-going assessment, active patient participation, and evidence of patient engagement in their care.'

7. How is living well measured in the 57 studies which had a quality of life outcome (EQ-5D is noted, what were the other measures)? It would be useful to know if the outcome of the primary studies was that patient perspectives of their wellbeing had improved (encompassing worldviews and perspectives of living well), or whether the primary outcome was improvement in patient function or reducing symptom burden (which may or may not have an impact on a person's perspective of their wellbeing). To make this clearer the outcome measure used for measuring QoL in each study should be included in the Table Supplementary material 4 (where possible). It is a shame there was not more data collected in the studies to inform health economic decisions.

Thank you for your interest in the outcome measures used within the primary studies to evaluate quality of life. To make this clearer, as suggested, we have added the outcome measures to the Supplementary Material 5 (previously Supplementary Material 4). As can be seen, most studies evaluating quality of life used participant self-report quality of life measures with fewer using objective measures of function. We agree it is unfortunate that the primary studies do not provide enough information to inform health economic analysis. We have amended the text as follows:

Service outcomes including costs

'Forty-five studies (58%) were included based on an effect on quality of life alone.

Fifty-seven studies (73%) used a disease or population specific tool to quantify quality of life (Supplementary Material 5) and five studies (6%) employed the Euro-Qual-5D (EQ-5D).'

8. Pg 15 line 56 add (n=stakeholders from LMIC)

We have amended the text as follows, including the number of stakeholders from LMIC as below:

'Stakeholders (n=20) contributed views from high-income countries (n=12) (UK, Japan, Taiwan, Portugal, Chile) and LMICs (n=8) (Uganda, Malawi, South Africa, Ghana, Zimbabwe, China, India and Bangladesh) contributed views.'

9. Correct typo on Supplementary material 6 (selective reporting heading).

Thank you for the opportunity to read and review your paper.

Thank you for spotting this, we have corrected it.

We hope that we have addressed the editorial and reviewers' comments to your satisfaction but are happy to provide further clarification if required.